# MM-Reasoner: A Multi-Modal Knowledge-Aware Framework for Knowledge-Based Visual Question Answering

**Mahmoud Khademi, Ziyi Yang, Felipe Vieira Frujeri, Chenguang Zhu**

Microsoft Cognitive Services Research Group

{mkhademi, ziyiyang, fevieira, chezhu}@microsoft.com

## Abstract

Thanks to the strong reasoning capabilities of Large Language Models (LLMs), recent approaches to knowledge-based visual question answering (KVQA) utilize LLMs with a global caption of an input image to answer a question. However, these approaches may miss key visual information that is not captured by the caption. Moreover, they cannot fully utilize the visual information required to answer the question. To address these issues, we introduce a new framework called Multi-Modal Knowledge-Aware Reasoner (MM-Reasoner) for KVQA. MM-Reasoner first utilizes a set of vision APIs, such as dense captioners, object detectors, and OCR, to extract detailed information from the image in textual format. Then, it prompts an LLM to extract query-specific knowledge from the extracted textual information to provide a rich representation that contains external knowledge, commonsense, explicit supporting facts, and rationales required for reasoning. Finally, the knowledge, query, and visual input are used to fine-tune a Vision-Language Model (VLM). At test time, MM-Reasoner uses the potential answers predicted by the VLM to iteratively update and optimize the prompt, refining its answer. Empirical studies show that MM-Reasoner achieves state-of-the-art performance on several KVQA datasets.

## 1 Introduction

The knowledge-based visual question answering (KVQA) task (Marino et al., 2019) requires not only visual information from the image, such as object attributes and visual relationship information, but also external knowledge, commonsense, rationales, and supporting facts for reasoning and predicting the correct answer. Therefore, a typical KVQA model consists of a knowledge retrieval module and an answer module. The traditional knowledge retrieval module usually retrieves knowledge from sources such as Wikipedia, knowledge graphs, and web search (Wu et al., 2022).

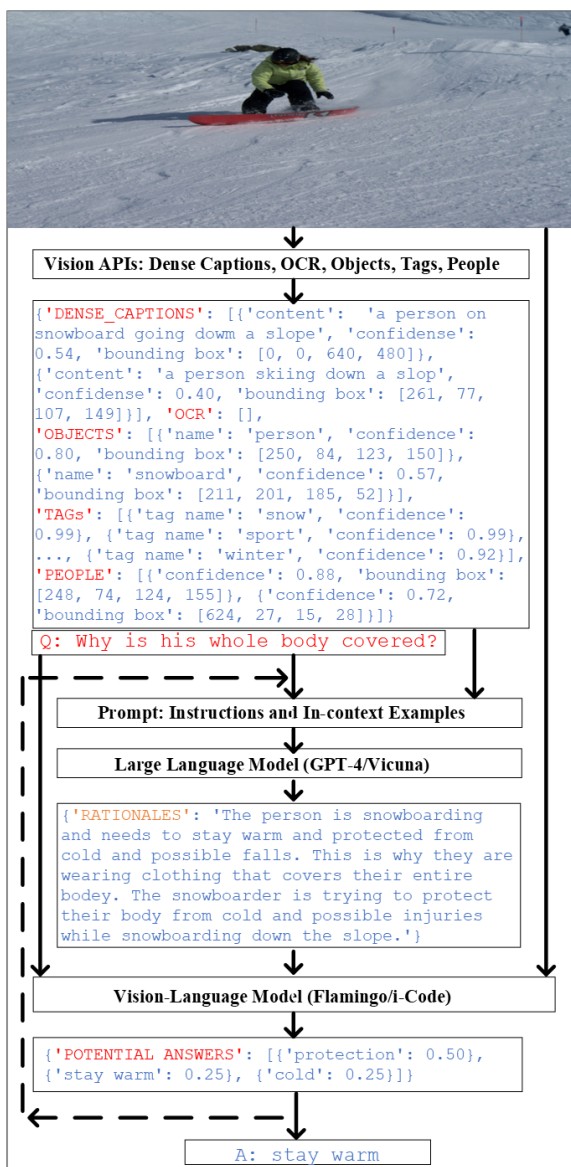

Figure 1: Multi-Modal Knowledge-Aware Reasoner

More recently, Large Language Models (LLMs) such as GPT-3 are used to produce related knowledge (Lin et al., 2022; Hu et al., 2022b). The latter approach is preferred since traditional knowledge retrieval often introduces irrelevant information to

the question (Yang et al., 2022a). After obtaining the required knowledge, the answer module takes the visual input or a global caption of the image, query, and knowledge to predict an answer. The answer module is either a transformer encoder-decoder architecture with trainable parameters (Lin et al., 2022) or more recently, a frozen LLM such as GPT-3 (Yang et al., 2022a; Hu et al., 2022a).

However, there are several key challenges to the above approaches. First, when using an LLM, the input needs to be text, but the recent approach of feeding the global caption of the input image into LLMs (Hu et al., 2022a) may miss key visual information. Second, the conceptual diversity of visual questions often necessitates multimodal reasoning with the integration of various external information beyond the image itself. These diverse types of knowledge include: commonsense, knowledge of concepts represented visually, external Knowledge, e.g., `Marco van Basten is a former football player born in the Netherlands.`, insights about the world's physics, e.g., `Mountain areas have a lower temperature than other areas`, etc. Third, recent approaches cannot fully utilize visual information in their answer module since they use an LLM to predict an answer. These approaches often struggle to answer visual questions that require fine-grained visual information, e.g., spatial relationships and object attributes, or knowledge of concepts represented visually, e.g., `What is the pattern of the table cloth called?`

To answer a diverse set of questions, a KVQA model requires (1) vision expertise, e.g., celebrity recognition, optical character recognition (OCR), and object detection (2) the reasoning capacity and world knowledge of the recent LLMs, and (3) the ability of Vision-Language Models (VLMs) to generate rich joint representations of vision and language. However, current knowledge-based VQA models cannot provide all these capabilities within a unified framework. To address these challenges, we introduce a new framework, called Multi-Modal Knowledge-Aware Reasoner (MM-Reasoner), for knowledge-based VQA (Figure 1). The MM-Reasoner first leverages a rich set of vision APIs/expertise such as dense captioner, object detector, tag detector, OCR, people detector and celebrity recognizer, to obtain a detailed textual representation of the visual input required to produce more accurate answers (Figure 2).

Then, the MM-Reasoner leverages an LLM to

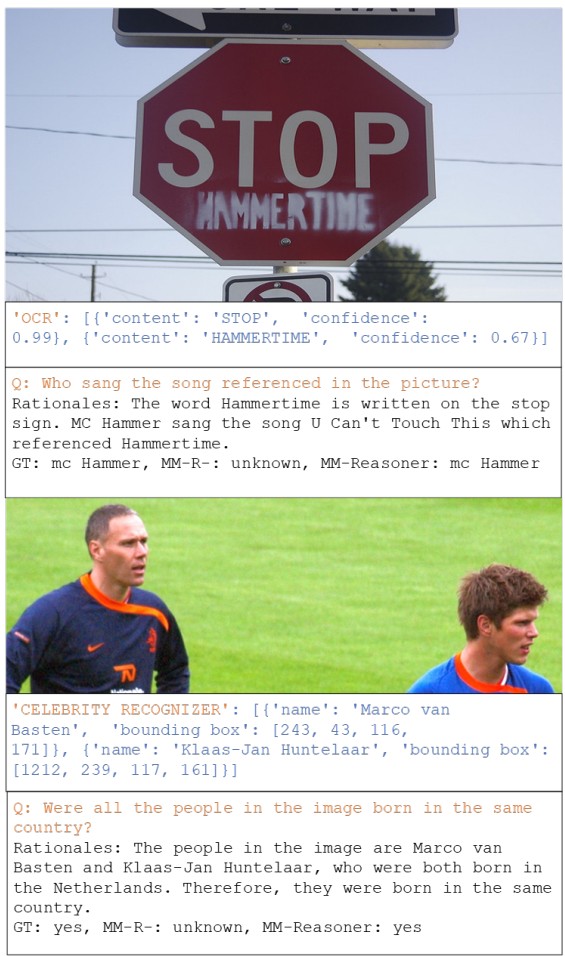

Figure 2: OCR and celebrity recognizer are helpful for answering questions. GT refers to the ground truth, and MM-R$^-$ is a baseline model similar to MM-Reasoner, but it uses a global caption instead of vision APIs.

extract query-specific knowledge from the textual representations in a self-supervised way via prompt designing, which requires only a few in-context examples. The LLM can provide a rich representation that contains external knowledge, commonsense, explicit supporting facts, and rationales required for reasoning. Finally, the extracted knowledge, the question, and the regional visual information of the input image are used to fine-tune a VLM. The VLM can fully utilize fine-grained visual information such as object regions, visual relationships/attributes, and visual spatial information supplementary to the textual representation to answer the question (see Figure 3). At test time, MM-Reasoner uses candidate answers predicted by the VLM to iteratively update and optimize the prompt to refine its final answer. This allows our framework to effectively exchange knowledge and visual information between the LLM and the VLM.

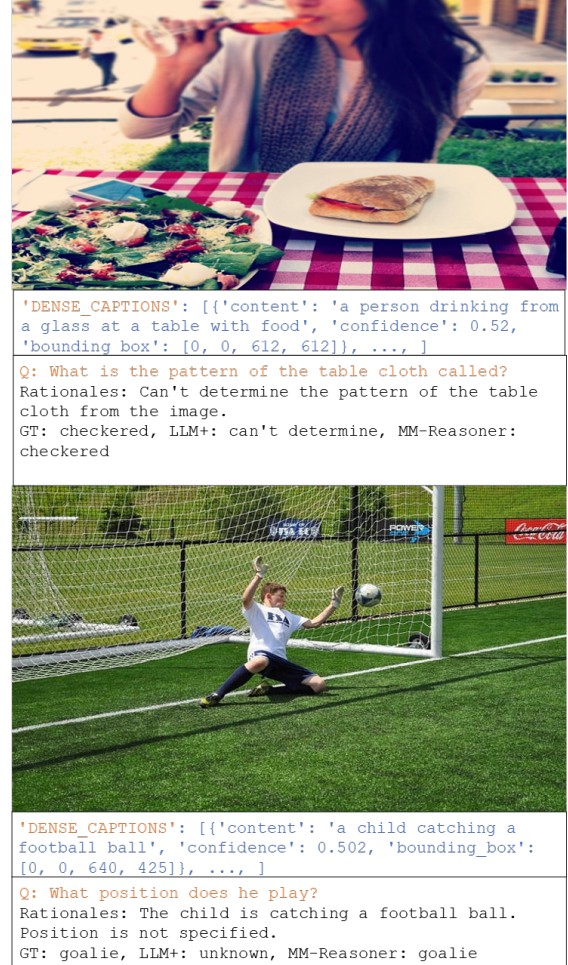

Figure 3: MM-Reasoner can effectively utilize visual features to find the correct answer by leveraging a VLM as its answer module. GT is ground truth and LLM[+] is a baseline model that prompts GPT-4 with textual descriptions obtained from vision APIs and the question to generate an answer. LLM[+] fails since the textual descriptions from APIs do not provide useful information.

MM-Reasoner is an integrative and composable framework that works with a variety of existing LLMs and VLMs. Since these LLMs and VLMs have already been trained on large-scale datasets, the fine-tuning cost of MM-Reasoner is much lower than that of existing models. We have conducted extensive empirical studies on MM-Reasoner using several datasets. The results show MM-Reasoner achieves state-of-the-art accuracy on the KVQA task, achieving 60.8% accuracy on the OK-VQA dataset, 60.2% on the A-OKVQA dataset, and 40.0% in the few-shot setting on the OK-VQA dataset. The project page is available at https://github.com/microsoft/i-Code/tree/main/mm-reasoner. In summary, the key contributions of this paper are as follows:

(1) We introduce a new framework for KVQA that offers vision APIs, the reasoning power of LLMs, and the rich joint representation of VLMs, all in a unified and composable framework.

(2) A key novelty of MM-Reasoner is that it effectively utilizes visual features in generating knowledge and predicting the final answer by leveraging a VLM and exchanging visual information and knowledge between the VLM and the LLM.

(3) The MM-Reasoner achieves state-of-the-art performance on several knowledge-based VQA benchmarks.

## 2 Related Work

Our work is related to several recent studies in the area of multi-modal learning and LLMs.

**Knowledge-Based VQA**. In REVIVE (Lin et al., 2022), the authors proposed to first employ an object detector to locate the objects, and then use the cropped bounding-box proposals to retrieve various types of external knowledge. Finally, they merged this knowledge with the regional visual features into a transformer to predict an answer. In Hu et al. (2022a), the authors introduced a captioning model called PromptCap, which is designed to generate question-aware captions using a GPT-3 prompt and question-answer pairs from existing VQA datasets. However, to generate the training examples, the PromptCap still relies on global captions of the image. Additionally, the visual features are not effectively utilized in the final answering model, as the answer module is a frozen LLM. In contrast, our prompt design enables our framework to extract query-specific knowledge from the visual input, avoiding the extraction of unrelated information. Additionally, MM-Reasoner harnesses the capabilities of VLMs such as Flamingo (Alayrac et al., 2022) and i-Code (Yang et al., 2022b) to incorporate visual features in the answer module.

In Hu et al. (2022b), the authors proposed an end-to-end model, called Retrieval-Augmented Visual Language Model (REVEAL), that learns to encode multimodal world knowledge into a memory and utilize it for addressing visual questions that demand extensive knowledge. REVEAL is composed of four modules: the memory, the encoder, the retriever, and the generator. The memory encodes different types of multimodal knowledge via a shared encoder. The retriever finds the most relevant information, and the generator uses that information with the input question to predict the

final answer. The model is pre-trained on a large amount of data and then fine-tuned for specific tasks. However, REVEAL does not leverage the reasoning capabilities of recent LLMs.

**Multimodal Composable Frameworks**. Recently, the composition of large pre-trained models has been extensively studied. The prevalent approach for composing these models is through joint fine-tuning on new tasks. Yang et al. (2022b) introduced i-Code, a self-supervised pretraining framework that allows users to merge vision, speech, and language modalities into comprehensive and versatile vector representations. In the i-Code framework, data from each modality is initially fed into pretrained single-modality encoders. The outputs from these encoders are subsequently combined using a multimodal fusion network that employs attention mechanisms and other novel architectural designs to efficiently integrate information across modalities. The i-Code is pretrained end-to-end using objectives such as masked modality unit modeling and cross-modality contrastive learning.

Wang et al. (2021) introduced a VLM framework, called SimVLM that is trained end-to-end using a single language modeling objective, simplifying the training process with large-scale weak supervision. Alayrac et al. (2022) proposed Flamingo, a set of VLMs capable of connecting pre-trained vision-only and language-only models, processing sequences of interleaved visual and textual data, and accepting images or videos as inputs. However, these methods can be computationally demanding. In contrast, models can be composed through a shared modality such as language. Zeng et al. (2022) proposed Socratic Models, a modular framework that allows multiple pre-trained models to exchange information, acquire new multimodal capabilities without fine-tuning.

**LLMs with Plugins and Tools**. Our framework is related to the use of tools and plugins for LLMs. In Toolformer (Schick et al., 2023), the authors demonstrated that language models can self-learn the utilization of external tools through simple APIs, such as calculators, search engines, and translators. Toolformer is designed to determine the appropriate APIs to invoke and the specific arguments to provide via a few in-context examples. In MM-REACT (Yang et al., 2023a), the authors integrated specialized vision APIs with ChatGPT to solve various visual understanding tasks. Through several demonstration examples, they showed that MM-REACT can effectively solve the given reasoning tasks, especially when tailored to the vision APIs. For instance, by employing an OCR API, it can gather information from multiple receipts and calculate the total cost. However, since MM-REACT relies solely on a textual representation of the input image and uses a frozen LLM to generate an answer, it often struggles to answer questions that require fine-grained visual information.

# 3 Multi-Modal Knowledge-Aware Reasoner

In this section, we present the Multi-Modal Knowledge-Aware Reasoner framework. As Figure 1 shows, MM-Reasoner is composed of three components: a set of vision APIs, an LLM, and a VLM. It is worth mentioning that MM-Reasoner is an integrative, modular, and composable framework that works with a variety of existing vision APIs, LLMs and VLMs. These components work together to extract intrinsic information from the image and query, extrinsic information from external knowledge, and conduct visual-textual reasoning to produce the answer. In the following, we explain these components in more details.

## 3.1 Vision APIs

The goal of vision APIs is to extract detailed visual information from the image in textual format, to be later used in LLM prompts. In our experiments, MM-Reasoner adopts Azure Cognitive Services for Vision (ACS Vision)[1]. The ACS Vision is a unified service that offers innovative computer vision capabilities. Our framework utilizes the following APIs from the Image Analysis SDK:

- Dense captions - Generates region-grounded captions for all important objects detected in an image.

- Tags - Extracts common tags from an image based on thousands of recognizable objects, living beings, scenery, and actions.

- Objects - Object detection is similar to tagging, but the API returns the bounding box coordinates for each tag applied.

- OCR - Extracts printed and handwritten text from an image.

---

[1] https://azure.microsoft.com/en-us/products/cognitive-services/vision-services

- People - Extracts bounding box coordinates of each detected person in an image.

- Celebrity Recognizer - Detects and identifies known celebrities in an image.

We aggregate the results from the Vision APIs to summarize the image with a set of captions, a set of objects/persons present in the image, a set of tags associated with the image, a set of texts written in the image, and the location of the detected people.

## 3.2 Large Language Model

Given the vision API outputs, we design a vision-informed language prompt to elicit rationales from LLM helpful to answer the question. The prompt is composed of instructions, a few in-context example instances, and the current input instance. Each instance includes an example question and the outputs of the vision APIs for the corresponding example image. Figure 4 shows a prompt that we use for the OK-VQA dataset. More specifically, given a question and visual input pair denoted by $(\mathbf{V}, \mathbf{Q})$, the input to the LLM is a prompt denoted by $\mathbf{P}$ obtained as follows:

$$\mathbf{P} = \mathsf{concat}\big(\mathbf{I}; (\mathbf{X}_{ic}; \mathbf{Y}_{ic})_1^k; \mathbf{X}\big) \qquad (1)$$

where, $\mathbf{I}$ denotes the instructions, $(\mathbf{X}_{ic}; \mathbf{Y}_{ic})_1^k$ are a list of $k$ in-context input-output pairs, and

$$\mathbf{X} = \mathsf{concat}\big(\mathsf{APIs}(\mathbf{V}); \mathbf{Q}\big) \qquad (2)$$

is obtained by appending the question $\mathbf{Q}$ to the outputs of the vision APIs for visual input $\mathbf{V}$. Inserting $k$ in-context input-output pairs before the $\mathbf{X}$ substantially improves the performance. With the prompt, the LLM utilize its strong reasoning capacity to respond with rationales. These rationales can be commonsense knowledge, external information, basic factual knowledge, supporting facts, or any information from the image that is required or helpful to answer the question.

## 3.3 Vision Language Model

Next, we feed the rationales, the image, and the question into the VLM. The VLM is fine-tuned on the training data to adapt to the rich input format. In this way, both the original visual information and external knowledge are leveraged to produce the answer. The framework supports both open-vocabulary and closed-vocabulary VLMs. The VLM's output is a list of potential answers with their confidence scores. For a closed-vocabulary VLM, this can be obtained by choosing a small subset of answers with the highest logits. For an open-vocabulary VLM, this list can be obtained using a beam search decoder or by fine-tuning multiple instances of the VLM and utilizing a greedy search decoder for each instance of the VLM. We use the latter method in our experiments. More specifically, we prompt the LLM $m$ times to get $m$ possibly different sets of rationales and fine-tune $m$ different instances of the VLM.

## 3.4 Further Improvements

**Iterative Reasoning.** At test time, we first use the above process to obtain a set of answers along with their confidence scores. The confidence score for each answer is acquired using the probability distribution over the predicted answers by the instances of the VLM. Instead of directly returning the answer with the highest confidence score as the final answer, we discovered that performance can be significantly improved by updating the prompt with the first pass prediction and repeating this process.

Specifically, we initially add answers with their confidence scores to the prompt and feed the updated prompt into the LLM. Then, we obtain a new set of rationales. These are often more informative than the previous rationales since the updated prompt provides VLM's answer suggestions. Next, we feed the new rationales into the instances of the VLM to acquire an updated set of potential answers. After repeating this process for a few iterations, we use the answer with the highest score as the final answer. We use two reasoning iterations after obtaining the initial answer. In our experiments, applying more iterations did not help to improve the performance. During the iterative reasoning, the instances of the VLM are kept frozen.

**k-NN-Based In-context Examples.** We apply a dynamic set of in-context examples during test time. First, we use a fine-tuned VLM to extract an embedding for each image and question pair in the dataset, utilizing the average encoder embedding. Then, for each test example, we select $k$ training examples that are closest to the test example based on the Euclidean distance between the example embeddings. These training examples are, in turn, used as in-context examples for this test example. The intuition is to use better in-context demonstrations by retrieving examples that are semantically similar to the test input (Liu et al., 2021).

```
The following is a conversation with an AI assistant. Given an image description and a question
about the image, the assistant  provides rationales that are required or helpful to answer the
question correctly. These rationales can be commonsense  knowledge, external information, basic
factual knowledge, supporting facts, or any information from the image that is required or
helpful to answer the question. Each image is described with a set of captions, a set of objects
present in the image, a set of tags associated with the image, a set of texts written in the
image, and the location of the people detected in the image. An empty list is represented by [].
The location of each caption, object, and detected person is specified by a bounding box. A
bounding box is a rectangle that surrounds an object, caption, or person in an image. Each
bounding box is represented by a list of 4 numbers as [x, y, w, h], where (x, y) corresponds to
the x and y coordinates of the top-left corner of the rectangle, w represents the width of the
bounding box, and h represents the height of the bounding box. The x, y, w, and h are normalized
to be between 0 and 1. So, (0, 0) and (1, 1) correspond to the x and y coordinates of the top-
left corner and bottom-right corner of the image, respectively. Each bounding box has a
confidence score that indicates how likely the object, caption, or person is actually present in
that bounding box. In each turn of the conversation, the human user describes an image and asks
a question about the image via a dictionary in the Python language. The question has a
unique id. The AI responds with a dictionary in the Python language. The dictionary has 3 keys:
"question id", "rationales", and "potential answers". If an image description lacks sufficient
information, the AI must still predict a list of potential answers to the question and provide
rationales.

Human: The description of the image and the given question are: {"captions": [{"caption": "a
snowboarder going down a slope", "confidence": 0.5, "bounding box": [0.0, 0.0, 1.0, 1.0]},
{"caption": "a snowboarder in the snow", "confidence": 0.5, "bounding box": [0.2, 0.1, 0.3,
0.4]}, {"caption": "a close-up of a person's foot", "confidence": 0.5, "bounding box": [0.2,
0.3, 0.1, 0.2]}], "objects": [{"object": "person", "confidence": 0.8, "bounding box": [0.2, 0.1,
0.3, 0.4]}], "tags": [{"tag": "snow", "confidence": 1.0}, {"tag": "sport", "confidence": 1.0},
{"tag": "outdoor", "confidence": 1.0}, {"tag": "person", "confidence": 1.0}, {"tag": "extreme
sport", "confidence": 0.9}, {"tag": "snowboard", "confidence": 0.9}, {"tag": "riding",
"confidence": 0.9}, {"tag": "piste", "confidence": 0.9}, {"tag": "winter sport", "confidence":
0.9}, {"tag": "adventure", "confidence": 0.9}, {"tag": "winter", "confidence": 0.9}, {"tag":
"slopestyle", "confidence": 0.9}, {"tag": "footwear", "confidence": 0.9}, {"tag": "glacial
landform", "confidence": 0.9}, {"tag": "recreation", "confidence": 0.8}, {"tag": "ski",
"confidence": 0.8}, {"tag": "freezing", "confidence": 0.8}, {"tag": "outdoor recreation",
"confidence": 0.8}, {"tag": "slope", "confidence": 0.8}, {"tag": "mountain", "confidence": 0.6},
{"tag": "snowboarding", "confidence": 0.6}, {"tag": "skiing", "confidence": 0.5}], "texts": [],
"people location": [{"confidence": 0.9, "bounding box": [0.2, 0.1, 0.3, 0.4]}], "question":
"Which season do you find this weather?", "question id": 42395}. Please provide the rationales
and a list of potential answers for the question.

AI: {"question id": 42395, "rationalels": "There is snow in the image. Weather is snowy
typically in Winter.", "potential answers": ["Winter"]}

.
.
.

Human: The description of the image and the given question are: {"caption": "horses running on a
hill", "confidence": 0.4, "bounding box": [0.0, 0.0, 1.0, 1.0]}, {"caption": "a horse running in
a field", "confidence": 0.4, "bounding box": [0.2, 0.4, 0.2, 0.2]}, {"caption": "a horse with a
large belly", "confidence": 0.3, "bounding box": [0.5, 0.4, 0.2, 0.2]}, {"caption": "a horse
with a tail", "confidence": 0.3, "bounding box": [0.4, 0.4, 0.1, 0.1]}, {"caption": "a horse
running in the grass", "confidence": 0.4, "bounding box": [0.8, 0.2, 0.2, 0.1]}], "objects":
[{"object": "mammal", "confidence": 0.8, "bounding box": [0.8, 0.3, 0.2, 0.1]}, {"object":
"mammal", "confidence": 0.8, "bounding box": [0.2, 0.4, 0.2, 0.1]}, {"object": "mammal",
"confidence": 0.6, "bounding box": [0.4, 0.4, 0.1, 0.1]}, {"object": "horse", "confidence": 0.6,
"bounding box": [0.5, 0.4, 0.2, 0.2]}], "tags": [{"tag": "outdoor", "confidence": 1.0}, {"tag":
"grass", "confidence": 1.0}, {"tag": "mammal", "confidence": 1.0}, {"tag": "animal",
"confidence": 1.0}, {"tag": "livestock", "confidence": 0.9}, {"tag": "field", "confidence":
0.9}, {"tag": "herd", "confidence": 0.9}, {"tag": "ranch", "confidence": 0.9}, {"tag":
"pasture", "confidence": 0.9}, {"tag": "sky", "confidence": 0.9}, {"tag": "herding",
"confidence": 0.9}, {"tag": "hill", "confidence": 0.8}, {"tag": "mountain", "confidence": 0.8},
{"tag": "landscape", "confidence": 0.7}, {"tag": "grazing", "confidence": 0.7}, {"tag":
"hillside", "confidence": 0.7}, {"tag": "horse", "confidence": 0.6}, {"tag": "grassy",
"confidence": 0.6}], "texts": [], "people location": [], "question": "What animals are these?",
"question id": "25EKstQLh3QXDC7R4Cggpr"}. Please provide the rationales and a list of potential
answers for the question.
```

Figure 4: The prompt used in MM-Reasoner for OK-VQA dataset. The black text provides instructions, while the blue and red texts demonstrate in-context examples and the current training example, respectively. At test time, we provide the potential answers predicted by the VML in addition to the image descriptions and question. The potential answers predicted by the LLM are not fed into the VLM. They only used for evaluating a baseline.

# 4 Experiments

This section covers the explanation of datasets and baseline models, followed by a discussion of experimental results and implementation details.

## 4.1 Datasets

We evaluate MM-Reasoner's performance on the following datasets: OK-VQA (Marino et al., 2019), A-OKVQA (Schwenk et al., 2022), FVQA (Wang et al., 2017), and KVQA (Shah et al., 2019). OK-VQA is a dataset for visual question answering that needs external knowledge to answer questions. OK-VQA includes 9,009 training questions and 5,046 test questions. A-OKVQA dataset is the augmented successor of OK-VQA. A-OKVQA questions are challenging, encompass a wide range of concepts, and unlike other knowledge-based visual question answering datasets, they cannot be addressed by merely consulting a knowledge base. The A-OKVQA offers rationales for answers in the training set. The rationales provide facts and snippets of reasoning required to answer the question. The train/validation/test split of the A-OKVQA dataset includes 17,056/1,145/6,702 questions. There is no overlap between question-image pairs in A-OKVQA and OK-VQA.

FVQA is a VQA dataset that mostly contains questions requiring external knowledge to answer, and provides supporting fact triplets alongside the image-question-answer triplets. Following (Wang et al., 2017), we used 1,090 test images, amounting to 2,899 questions. The KVQA (Shah et al., 2019) dataset contains 183,007 pairs of questions and answers, which involve 18,880 named entities and 24,602 images. The questions in this dataset require complex reasoning over large knowledge graphs, involving multiple entities, relations, and hops to arrive at an answer. KVQA is currently one of the largest datasets available for studying VQA over knowledge graphs. KVQA has 5 different train/validation/test splits. We use the last one.

## 4.2 Baselines

We compared our framework with several models that have recently been developed, including the state-of-the-art models PromptCap, Flamingo, and REVIVE. For further comparison, we also include two models that were proposed more recently in Arxiv preprints during the preparation of this work: Prophet and REVEAL. Additionally, to verify the significance of each component in our framework,

we performed an ablation study by implementing five baseline models: $LLM^+$, $VLM^+$, $MM-R^-$, $MM-R^\circ$, and MM-R (see Table 1).

In $LLM^+$, the VLM component is removed. Hence, $LLM^+$ does not use the image directly to generate an answer. This baseline is designed to study the significance of the VLM module in our framework. A prompt is constructed using the vision APIs and the given question. Each in-context example is modified so that GPT-4 provides a single answer to the given question in addition to the rationales. For each example in the test set, the prompt is fed into the GPT-4. Then, the predicted answer is mapped to one of the words in the test set vocabulary based on the cosine similarity between the embedding of the predicted answer and the embedding of each word in the vocabulary. We use this method because the LLM's answers often differ from the ground truth, even when an answer is semantically correct. We use GloVe (Pennington et al., 2014) to obtain the embedding for the predicted answer and each word in the vocabulary. No fine-tuning is applied to $LLM^+$.

| Baseline | APIs | LLM | VLM | IT-R | Acc |
|----------|------|-----|-----|------|-----|
| $LLM^+$ | √ | √ | × | × | 38.8 |
| $VLM^+$ | √ | × | √ | × | 53.8 |
| $MM-R^-$ | × | √ | √ | × | 57.1 |
| $MM-R^\circ$ | √ | √ | √ | × | 58.5 |
| MM-R | √ | √ | √ | √ | **59.2** |

Table 1: Ablation study on OK-VQA dataset. IT-R denotes iterative reasoning.

In $VLM^+$, the LLM module is removed. The input to the i-Code v2 is the output of the vision APIs, the image, and the given question. $VLM^+$ does not exploit the reasoning capabilities of the LLM module, and its external knowledge is limited to the VLM pre-trained knowledge only. $VLM^+$ is fine-tuned with the ground truth answers. MM-R is a single MM-Reasoner framework without leveraging ensemble learning. $MM-R^\circ$ is similar to the MM-R, except that at test time, the output of the VLM is used as the final answer. That is, the VLM does not update the initial prompt. This baseline is designed to study the significance of the iterative reasoning in our framework. Finally, the $MM-R^-$ is similar to $MM-R^\circ$, except that we only use a global image caption and the question to construct the prompt. This baseline is designed to study the

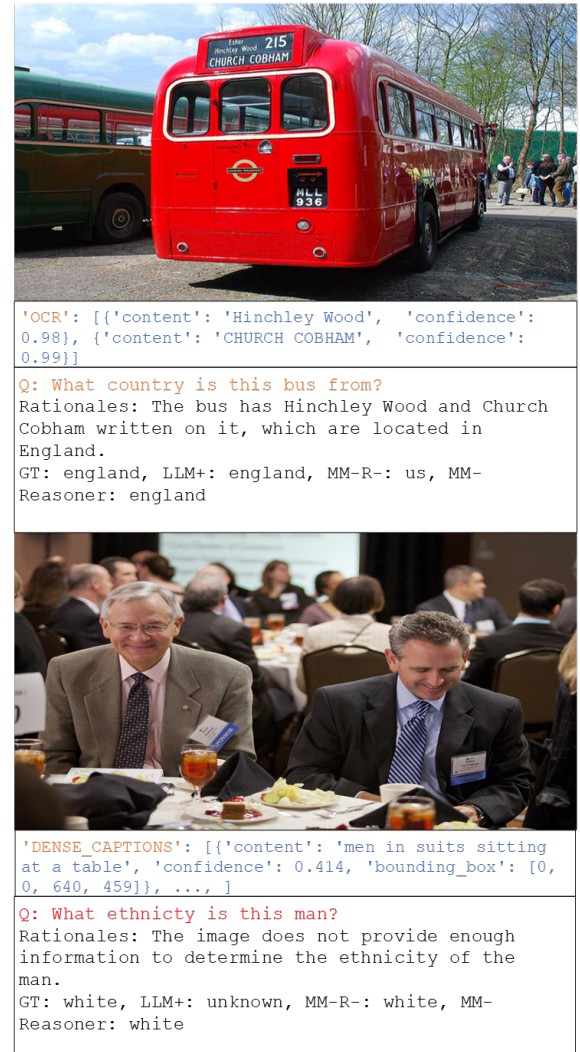

```
'OCR': [{'content': 'Hinchley Wood',  'confidence':
0.98}, {'content': 'CHURCH COBHAM', 'confidence':
0.99}]
```

Q: What country is this bus from?
Rationales: The bus has Hinchley Wood and Church
Cobham written on it, which are located in
England.
GT: england, LLM+: england, MM-R-: us, MM-
Reasoner: england

```
'DENSE_CAPTIONS': [{'content': 'men in suits sitting
at a table', 'confidence': 0.414, 'bounding_box': [0,
0, 640, 459]}, ..., ]
```

Q: What ethnicty is this man?
Rationales: The image does not provide enough
information to determine the ethnicity of the
man.
GT: white, LLM+: unknown, MM-R-: white, MM-
Reasoner: white

Figure 5: VQA examples on OK-VQA test set. GT refers to ground truth. For LLM$^+$ and MM-R$^-$, see 4.2.

significance of vision APIs in our framework.

### 4.3 Results and Discussion

Our results on OK-VQA and A-OKVQA datasets are shown in Table 2 and Table 3, respectively. MM-Reasoner outperforms all baseline models, including the state-of-the-art models PromptCap, Flamingo, and REVIVE on both datasets. Moreover, in the few-shot setting, MM-Reasoner significantly surpasses the state-of-the-art Flamingo-9B by a considerable difference of 4.3 points (Table 4). This demonstrates that MM-Reasoner can integrate the strengths of both LLMs and VLMs to answer complex visual questions. Figure 5 shows two VQA examples. MM-Reasoner can find the answer to the first challenging question using OCR API. In the second example, LLM$^+$, which leverages GPT-4 and vision APIs, cannot find the answer since

the question requires fine-grained visual information not described by the APIs. However, MM-Reasoner can find the answer by leveraging the power of its VLM. Figure 6 shows our framework can refine its answer via the iterative reasoning.

The results of our ablation study are summarized in Table 1. LLM$^+$ performs poorly because the textual representation of the image alone does not provide sufficient fine-grained visual information for reasoning. MM-R$^-$ outperforms both LLM$^+$ and VLM$^+$, as it can leverage the power of both LLMs and VLMs. Additionally, MM-R$^\circ$ outperforms MM-R$^-$, demonstrating that vision APIs can help improve performance. Lastly, MM-R surpasses MM-R$^\circ$, indicating that iterative reasoning can contribute to enhanced performance. We present our findings on the FVQA dataset in Table 5. Notably, FVQA (Wang et al., 2017) utilized the ground truth supporting fact triplets included in the dataset. In contrast, MM-Reasoner surpasses this baseline without relying on these triplets. Furthermore, our framework outperforms the Mem-Net implementation by Shah et al. (2019), which achieved a 59.2 accuracy on the KVQA test set, by +2.2 points, resulting in a 61.4 accuracy. It is important to note that MemNet uses ground truth entity names to recognize celebrities in KVQA, whereas we employ an API for this task.

| Model | Acc |
|---|---|
| PICa (Yang et al., 2022a) | 48.0 |
| KAT (Ensemble) (Gui et al., 2021) | 54.4 |
| Unified-io (2.8B) (Lu et al., 2022) | 54.0 |
| REVIVE (Lin et al., 2022) | 56.6 |
| Flamingo (80B) (Alayrac et al., 2022) | 57.8 |
| REVIVE (Ensemble) (Lin et al., 2022) | 58.0 |
| REVEAL-Large (Hu et al., 2022b) | 58.0 |
| REVEAL (Hu et al., 2022b) | 59.1 |
| PromptCap (Hu et al., 2022a) | 60.4 |
| MM-Reasoner | 59.2 |
| MM-Reasoner (Ensemble) | **60.8** |

Table 2: Performance on OK-VQA test set.

### 4.4 Implementation Details

GPT-4-32k and Vicuna-13B (Chiang et al., 2023) from FastChat[2] are used as the LLM in our framework. GPT-4-32k extends the context-length to 32,000 tokens compared to GPT-4. The maximum

[2]https://github.com/lm-sys/FastChat

| Model | Acc |
|---|---|
| ClipCap (Schwenk et al., 2022) | 25.9 |
| ViLBERT (Schwenk et al., 2022) | 25.9 |
| LXMERT (Schwenk et al., 2022) | 25.9 |
| KRISP (Schwenk et al., 2022) | 27.1 |
| GPV-2 (Schwenk et al., 2022) | 40.7 |
| Unified-IO (Lu et al., 2022) | 45.2 |
| REVEAL (Hu et al., 2022b) | 52.2 |
| Prophet (Shao et al., 2023) | 55.7 |
| PromptCap (Hu et al., 2022a) | 59.6 |
| MM-Reasoner (Ensemble) | **60.2** |

Table 3: Performance on A-OKVQA test set.

token context-length for Vicuna-13B is 2,048, but it is much faster than GPT-4. We use GPT-4-32k for experiments on OK-VQA, A-OKVQA, and FVQA, while Vicuna-13B is used for experiments on the KVQA dataset since it is significantly larger than other knowledge-based VQA datasets. The LLM module remains frozen in all of our experiments. The `temperature` and `top_p` are both set to 0.98. Additionally, `frequency_penalty` and `presence_penalty` are both set to 0. The number of in-context examples, $k$, is set to 20 for GPT4-32k and 2 for Vicuna. We utilize APIs from the Image Analysis SDK version v4.0, except for the celebrity recognizer. The celebrity recognizer API is used for the KVQA dataset only.

We use i-Code v2 (Yang et al., 2023b)[3] pre-finetuned on the VQA v2.0 dataset (Goyal et al., 2017) and Flamingo-9B (Alayrac et al., 2022) from OpenFlamingo (Awadalla et al., 2023) [4] as the VLM in our framework. We utilize Flamingo-9B only for few-shot experiments, while i-Code v2 is used in all other experiments. The i-Code v2 is fine-tuned during training, while Flamingo-9B remains frozen. The i-Code v2 prompt is "Answer the following question based on the given image: `<Question>` Hint: `<Rationales>`". Note that we use an open-vocabulary VLM. All fine-tuning jobs are conducted on two RTX A6000 GPUs in less than a few hours. The learning rate, batch size, and number of epochs are set to $2.0 \times 10^{-6}$, 4, and 8, respectively. The number of VLM instances, $m$, is set to 4. For ensemble learning, we fine-tune 32 instances of the entire framework with different

[3] https://github.com/microsoft/i-Code
[4] https://github.com/mlfoundations/open_flamingo

initialization and select the final answer based on majority voting. For Flamingo-9B, 8 shots is used and cross attention is applied after every 4 layers.

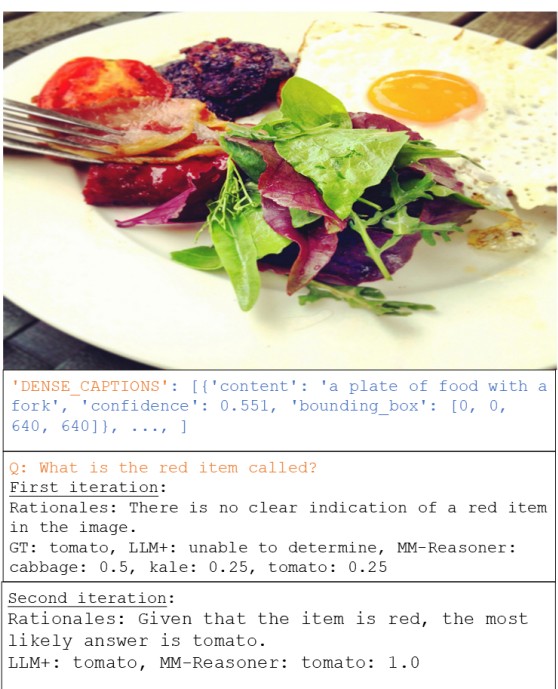

```
'DENSE_CAPTIONS': [{'content': 'a plate of food with a
fork', 'confidence': 0.551, 'bounding_box': [0, 0,
640, 640]}, ..., ]

Q: What is the red item called?
First iteration:
Rationales: There is no clear indication of a red item
in the image.
GT: tomato, LLM+: unable to determine, MM-Reasoner:
cabbage: 0.5, kale: 0.25, tomato: 0.25

Second iteration:
Rationales: Given that the item is red, the most
likely answer is tomato.
LLM+: tomato, MM-Reasoner: tomato: 1.0
```

Figure 6: MM-Reasoner can refine its answer via iterative reasoning. In the second iteration, the VLM answers are added to the prompt and fed into the LLM to generate updated rationales for the VLM.

| Method | Acc |
|---|---|
| Flamingo-9B (Alayrac et al., 2022) | 35.7 |
| MM-R° | **40.0** |

Table 4: Few-shot performance on OK-VQA.

| Method | Acc |
|---|---|
| Human | 77.99 |
| FVQA (Wang et al., 2017) | 56.91 |
| ZS-VQA (Chen et al., 2021) | 58.27 |
| FVQA (Ensemble) (Wang et al., 2017) | 58.76 |
| MM-Reasoner (Ensemble) | **61.1** |

Table 5: Performance on FVQA dataset.

## 5 Conclusion

MM-Reasoner is a new, unified framework for KVQA that combines vision APIs, the capabilities of LLMs, and VLMs. Experiments showed MM-Reasoner is an effective framework for KVQA.

## Limitations

Although the MM-Reasoner framework demonstrates strong performance in knowledge-based visual question answering tasks, it has some limitations worth mentioning:

**Frozen LLM:** The LLM used in MM-Reasoner remain frozen throughout the experiments, meaning it is not fine-tuned during training. This might limit the LLM's potential to adapt to the specific downstream tasks or evolving datasets.

**Leveraging Language APIs and Tools:** The MM-Reasoner framework can be extended to incorporate language APIs and tools, such as search, translator, and calendar, to answer questions that require up-to-date information or time-related context. By integrating these APIs and tools, the framework can access the latest information and adapt its reasoning process accordingly. For example, consider the following QA scenarios that require the latest information:

*Question:* `Who won the most recent edition of the tournament in the image?` A search API can be useful here to retrieve the latest winner of the tournament, as this information might not be present in the LLM's pre-existing knowledge.

*Question:* `When is the next public holiday represented in the image?` A calendar API can provide the necessary information about upcoming public holidays, allowing the framework to generate an accurate answer based on the current date and the holiday calendar of the given country. By integrating such language APIs and tools, MM-Reasoner can enhance its ability to handle a broader range of questions that require up-to-date information or specific time-related context, resulting in improved performance in knowledge-based visual question answering task.

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
