# OpenReview forum: "MM-Reasoner: A Multi-Modal Knowledge-Aware Framework for Knowledge-Based Visual Question Answering"
_EMNLP/2023/Conference — EMNLP 2023 Findings_

### Official Review · Reviewer_tUjL · 2023-08-05

**Soundness:** 3

**Excitement:**

4: Strong: This paper deepens the understanding of some phenomenon or lowers the barriers to an existing research direction.

**Missing References:**

Null

**Paper Topic And Main Contributions:**

This paper presents MM-Reasoner, a new framework for knowledge-based visual question answering (KVQA) tasks. The system combines the strengths of large language models (LLMs), vision APIs, and vision-and-language models (VLMs) to handle complex questions that require external knowledge and fine-grained visual understanding.

**Questions For The Authors:**

1. Why do we need to use an additional vision language model? There are two heavy large language models used in this work. And I think it makes no sense to need two complex models to do duplicated things.

**Reasons To Accept:**

1. They consider the VQA problem in the ChatGPT area
2. They used the visual recognized result to enhance the context information with a generative model

**Reasons To Reject:**

1. The overall framework is heavy and not that makes sense.
2. The text and formatting needs to be improved.

**Reproducibility:**

3: Could reproduce the results with some difficulty. The settings of parameters are underspecified or subjectively determined; the training/evaluation data are not widely available.

**Reviewer Confidence:**

3: Pretty sure, but there's a chance I missed something. Although I have a good feel for this area in general, I did not carefully check the paper's details, e.g., the math, experimental design, or novelty.

---

> ### Author Rebuttal · Authors · 2023-08-29
>
> Thank you for your insightful comments. As we discussed in the introduction, the KVQA task involves a diverse set of questions, some requiring extensive external knowledge, language understanding, and reasoning capabilities, while others necessitate understanding visual concepts such as the checkered pattern of a table cloth. To address this diversity, our proposed MM-Reasoner aims to provide (1) vision expertise, e.g., celebrity recognition, (2) the reasoning capacity and world knowledge of recent LLMs, and (3) the ability of VLMs to generate rich joint representations of vision and language within a unified framework. This approach allows our model to outperform state-of-the-art performance on various diverse datasets. Another key motivation to use both LLMs and VLMs is that LLMs are trained using text data like Wikipedia and books, whereas VLMs utilize image-caption pairs for training. This distinction in pre-training domains results in differing, and occasionally complementary, abilities for LLMs and VLMs. For example, LLMs are well-suited for questions requiring reasoning capabilities, but they have difficulty understanding visual patterns. MM-Reasoner capitalizes on the strengths of both LLMs and VLMs (as noted by Reviewer wKib) to tackle the KVQA task.
>
> An alternative method would be to train a model with massive datasets of image-text pairs plus massive text-only datasets, as in GPT-V (GPT-4 with visual input). However, this approach is more expensive, and the model may not be easily accessible. Furthermore, the concept of adding tools and plugins remains valuable for models like GPT-V, as it increases the capabilities of the model and reduces the number of training examples required to train the model. Experiments show that GPT-V sometimes fails to recognize celebrities, for instance.
>
> In comparison to state-of-the-art frameworks that utilize LLMs like GPT-3 and require highly expensive training and evaluation processes, MM-Reasoner offers a more cost-effective solution. Its modular and integrative framework is compatible with a wide range of existing APIs, LLMs, and VLMs, including open-source options, Hugging Face models, and Azure Cognitive Services/APIs. These LLMs/VLMs have already been trained on extensive datasets, making the fine-tuning process for MM-Reasoner significantly less costly. In fact, all fine-tuning tasks can be completed on two RTX A6000 GPUs in just a few hours (Line 541).
>
> Moreover, MM-Reasoner's modular and composable architecture allows it to benefit from advancements in knowledge distillation methods, which can reduce the model size of LLMs and VLMs and further enhance the efficiency of our proposed framework.
>
> Regarding the format: We will improve the format and move Figure 6 to section 3, as the extra page allows for it.

---

### Official Review · Reviewer_XP8j · 2023-08-05

**Soundness:** 4

**Excitement:**

4: Strong: This paper deepens the understanding of some phenomenon or lowers the barriers to an existing research direction.

**Paper Topic And Main Contributions:**

This paper proposes a new framework called Multi-Modal Knowledge-Aware Reasoner (MM-Reasoner) for KVQA, which utilizes a set of vision APIs to extract detailed information from the image in textual format and then prompts an LLM to extract query-specific knowledge from the extracted textual information to provide a rich representation. Finally, the knowledge, query, and visual input are used to fine-tune a Vision-Language Model (VLM). Empirical studies show that MM-Reasoner achieves state-of-the-art performance on several KVQA datasets.

**Questions For The Authors:**

In Eq 1, how is Y_ic obtained?
In Eq 2, what does V and Q denote?

**Reasons To Accept:**

1. The method is novel in utilizing vision APIs to extract various features and inputting this information into an LLM to get query-specific knowledge. This operation makes a good use of the LLM and can obtain richer information, external knowledge, commonsense, explicit supporting facts, and rationales.
2. Experimental results show that the proposed method outperforms current approaches and achieves SOTA performance.


**Reasons To Reject:**

1. The framework is quite a big one which uses many external APIs and an LLM, which may make it difficult to deploy. And what is the cost and efficiency?
2. Some details in the process of prompt construction may be made clearer. For example, how are the examples selected or constructed? How many examples are used for each sample? Also, I think Figure 6 is very important for understanding the method, however, it is in the Appendix due to page limit and may be not seen by the readers when the paper is published. Is it possible to add some key information of Figure 6 into the paper?


**Reproducibility:**

3: Could reproduce the results with some difficulty. The settings of parameters are underspecified or subjectively determined; the training/evaluation data are not widely available.

**Reviewer Confidence:**

3: Pretty sure, but there's a chance I missed something. Although I have a good feel for this area in general, I did not carefully check the paper's details, e.g., the math, experimental design, or novelty.

---

> ### Author Rebuttal · Authors · 2023-08-29
>
> Thank you for your insightful comments and suggestions. Regarding the cost and efficiency, all state-of-the-art frameworks utilize LLMs, such as GPT-3, and train with vast knowledge resources and VQA datasets to address this challenging task. Consequently, both the training and evaluation processes are highly expensive in these frameworks. In contrast, MM-Reasoner, as a modular and integrative framework, is compatible with a wide range of existing APIs, LLMs, and VLMs, including open-source options, Hugging Face models, and Azure Cognitive Services/APIs. Since these LLMs/VLMs have already been trained on extensive datasets, the fine-tuning process for MM-Reasoner is significantly less costly compared to other cutting-edge approaches. In fact, all fine-tuning tasks can be accomplished on two RTX A6000 GPUs in just a few hours (Line 541). Additionally, the modular and composable architecture of MM-Reasoner allows it to benefit from advancements in knowledge distillation methods. These methods can reduce the model size of LLMs and VLMs, further enhancing the efficiency of MM-Reasoner.
>
> Regarding the prompt construction, we appreciate the opportunity to clarify the details of our prompt construction process. In-context examples are selected from the training data, with the A-OKVQA dataset providing ground-truth rationales for answers in the training set (Line 393). For the OK-VQA dataset, we manually created rationales for a small subset of in-context examples, which will be made available on the MM-Reasoner Github page. This small subset serves as the basis for our dynamic in-context example set, tailored to each image and question pair as described in the k-NN-based in-context examples subsection (Line 362).
>
> The number of in-context examples, k, is set to 20 for GPT-4-32k and 2 for Vicuna (Line 525). For GPT-4-32k, the maximum token context-length is 32,000, while for Vicuna-13B, it is limited to 2,048 (Line 514). Due to this constraint, Vicuna-13B can accommodate only 2 in-context examples.
>
> Regarding Figure 6: We appreciate your suggestion and will move Figure 6 to Section 3 in the final version, as the extra page allows for it. Additionally, the prompts will be shared on the MM-Reasoner Github page.
>
> For Eq 1, the ground-truth rationale for the in-context example X_ic, denoted by Y_ic, is obtained as follows - for the A-OKVQA dataset, the Y_ic is provided by the creators of the dataset for each training and validation samples (Line 393). For the OK-VQA dataset, we manually created the ground-truth rationales for a small subset of in-context examples. An example of a rationale for the OK-VQA dataset can be found in Figure 6. We will make these rationales available on the MM-Reasoner GitHub page. As for Eq 2, Q represents the input question, while V denotes the given visual input or image (Line 299).

---

### Official Review · Reviewer_wKib · 2023-08-06

**Soundness:** 3

**Excitement:**

4: Strong: This paper deepens the understanding of some phenomenon or lowers the barriers to an existing research direction.

**Paper Topic And Main Contributions:**

* This paper proposes a multi-modal knowledge-aware framework (MM-Reasoner) exploiting a large language model and a vision-language model. MM-Reasoner consists of a set of vision APIs, an LLM, and a VLM, and these components cooperate to predict an answer to a given question by extracting and exchanging the intrinsic and extrinsic knowledge about a given image and the question.

**Questions For The Authors:**

- Question A: The number of in-context examples is 20 for GPT-4 and 2 for Vicuna. How did you set the number of examples? The variance of the number seems to be high.
- Question B: This paper adopts 4 number of VLM instances (described in L545). Have you conducted an ablation study on the number of VLM instances? Multiple VLM instances are necessary?

**Reasons To Accept:**

- The topic discussed in this paper and the idea of the proposed approach leveraging LLM, VLM, and additional plugins would draw attention in the NLP community and well suits the EMNLP conference.
- The idea of this paper is simple but meaningful. It explores not only LLMs with the plugins but also cooperation between an LLM and a VLM. In addition, further improvements in iterative reasoning and similarity-based in-context learning also be interesting and would promote further related studies.

**Reasons To Reject:**

- The experimental results are somehow weak. The performance improvement seems to be marginal. Especially, the only ensemble results are reported in Tables 2, 3, and 5. The result of using a single model should be reported for a fair comparison.
- The details about the few-shot experiment are almost missing. The motivation and the implementation details should be explained.

**Reproducibility:**

3: Could reproduce the results with some difficulty. The settings of parameters are underspecified or subjectively determined; the training/evaluation data are not widely available.

**Reviewer Confidence:**

4: Quite sure. I tried to check the important points carefully. It's unlikely, though conceivable, that I missed something that should affect my ratings.

---

> ### Author Rebuttal · Authors · 2023-08-29
>
> Thank you for your valuable comments. Regarding the experimental results, we have indeed reported the performance of a single model (MM-R) in Table 1 (Acc=59.2). MM-R represents a single MM-Reasoner framework without the use of ensemble learning (as mentioned in Line 453). It is important to note that our single model performance surpasses that of state-of-the-art models such as REVIVE (Ensemble), Flamingo (80B), REVEAL-Large, and REVEAL, as shown in Table 2. To facilitate a more straightforward comparison, we will add the MM-R single performance to Table 2 as you suggested. Also, in the case of few-shot experiments, we employed another single MM-Reasoner framework (MM-R^o), which significantly outperforms Flamingo (80B) as presented in Table 4.
>
> For the few-shot experiments, Flamingo-9B was utilized (referenced in Line 535). We employed 8 shots for Flamingo-9B and applied cross attention after every 4 layers (as stated in Line 549). All other hyperparameters were set to the default values provided in the open_flamingo Github repository (https://github.com/mlfoundations/open_flamingo). We will ensure that these details are clearly articulated in the paper and will also include them on the MM-Reasoner Github page.
>
> In response to Question A regarding the varying number of in-context examples for GPT-4 and Vicuna: The primary reason for setting the number of in-context examples is based on the observation that performance often improves when the number of in-context examples increases. Consequently, we determined the number of in-context examples according to the maximum token context-length permitted by each model. For GPT-4-32k, the maximum token context-length is 32,000, while for Vicuna-13B, it is limited to 2,048 (as stated in Line 514). Due to this constraint, Vicuna-13B can only accommodate 2 in-context examples.
>
> In response to Question B regarding the use of 4 VLM instances: The primary motivation for employing multiple VLM instances is to facilitate Iterative Reasoning by examining a range of candidate answers instead of solely focusing on the highest-confidence response. While utilizing 4 VLM instances (m=4) is one approach to achieve this, alternative methods such as incorporating a beam search decoder (referenced in Line 327) could also be employed. Comparative analysis revealed that using 4 VLM instances improved performance by approximately 0.7 points compared to a single VLM instance (m=1). However, increasing the number of instances beyond 4 (m>4) did not yield any further improvements.

---

### Meta-Review · Area_Chair_u8Vo · 2023-09-19

**Recommendation:** 4

**Metareview:**

The paper introduces MM-Reasoner, a framework designed for knowledge-based Visual Question Answering (VQA) that harnesses the strengths of multiple models and tools. Specifically, when presented with an image and a question, MM-Reasoner engages vision APIs, an LLM, and a vision-language model in tandem to address the query.

All reviewers commend the innovative approach of amalgamating the capabilities of various models and plugins. The method demonstrates commendable empirical results on the OKVQA benchmark. However, some reservations exist. Two reviewers note that the framework is intricate, necessitating intensive engineering of the pipeline. They also highlight concerns about the assumed availability and access to a multitude of tools/plugins, such as APIs.

---

### Decision · Program_Chairs · 2023-10-07

**Decision:**

Accept-Findings

**Comment:**

The paper introduces MM-Reasoner, a framework designed for knowledge-based Visual Question Answering (VQA) that harnesses the strengths of multiple models and tools. Specifically, when presented with an image and a question, MM-Reasoner engages vision APIs, an LLM, and a vision-language model in tandem to address the query.

All reviewers commend the innovative approach of amalgamating the capabilities of various models and plugins. The method demonstrates commendable empirical results on the OKVQA benchmark. However, some reservations exist. Two reviewers note that the framework is intricate, necessitating intensive engineering of the pipeline. They also highlight concerns about the assumed availability and access to a multitude of tools/plugins, such as APIs.